# Unveiling the Predisposing Factors for the Development of Branch Canker and Dieback in Avocado: A Case of Study in Chilean Orchards

Ana L. Valencia [1], Jorge Saavedra-Torrico [2], Inés Marlene Rosales [1], Johanna Mártiz [1], Alonso Retamales [1], Andrés Link [3] and Pilar M. Gil [1],*

[1] Facultad de Agronomía e Ingeniería Forestal, Pontificia Universidad Católica de Chile, Casilla 306-22, Santiago 7810000, Chile
[2] Escuela de Ingeniería de Alimentos, Facultad de Ciencias Agronómicas y de los Alimentos, Pontificia Universidad Católica de Valparaíso, Av. Brasil 2950, Valparaíso 2340000, Chile
[3] Agricom—Westfalia Chile, El Golf 99, Piso 3 Las Condes, Santiago 7550000, Chile
* Correspondence: pmgil@uc.cl

**Abstract:** Chilean avocado orchards have been affected by branch canker and dieback, mainly associated with pathogens species of the family Botryosphaeriaceae. Producers often relate water stress to these phenomena; however, predisposing factors for these diseases in Chilean orchards have not been reported. Therefore, the objective of this study was to identify the main climate, planting, and management variables associated with branch canker and dieback in Chilean avocado orchards. Multivariable statistical analyses (PCA and PLS-DA) were performed to analyze 76 variables possibly associated with the incidence of these diseases during two consecutive growing seasons. Our findings indicated that branch canker and dieback are mainly conditioned by orchard variables such as plant age, canopy volume, trunk diameter, leaf area index, and planting density. Variables associated with aged orchards, such as high age, high canopy volume, and high trunk diameter, appeared to be more associated with high incidence levels than were younger orchards with higher planting densities and high tree vigor levels. In addition, abiotic stresses, such as frost and overirrigation, were associated as secondary factors affecting these diseases' development in avocado orchards. Moreover, this study allowed us to determine that branch canker and dieback can develop indistinctly in the different Chilean agroclimatic zones.

**Keywords:** *Persea americana*; Botryosphaeriaceae; incidence; wood diseases; multivariate analysis

## 1. Introduction

Avocado (*Persea americana* Miller) is a perennial species endemic to Central America and Mexico. Avocados are produced mainly in Mexico, Chile, the Dominican Republic, Peru, Indonesia, Colombia, Brazil, Kenya, the USA, Venezuela, South Africa, Australia, New Zealand, and Israel [1]. Chile is an important avocado producer and exporter, with 29,224 ha planted between the Coquimbo Region (29°20′ S to 32°15′ S) and the O'Higgins Region (33°51′ S to 35°01′ S), with the Valparaiso Region being the most productive area [2].

"Hass" is the main avocado cultivar produced in Chile and is massively exported to Europe, North America, Asia, and South America. The avocado growing area is concentrated in the central zone of the country, which has a Mediterranean climate (semidesert and temperate); most of the new plantations are situated on hill slopes to avoid winter frosts, which can severely affect avocado production [3]. In general, avocado production in Chile has low disease pressure [4]; nevertheless, increasing cases of branch canker and dieback have been reported in avocado orchards since 2011 [5], coinciding with a severe drought affecting the avocado production area in the last thirteen years that caused a significant reduction in the planted area [2]. Water stress in avocado trees causes physiological

disorders and damage to the trees and allows latent pathogens to cause wood diseases, both via inner tissue and new colonization [6,7].

Branch canker and dieback are two woody plant diseases mainly associated with a complex of fungal species, of which the most common fungal pathogens on avocado are species of the family Botryosphaeriaceae [7,8]. In avocado-producing countries worldwide, the Botryosphaeriaceae species associated with these diseases are *Botryosphaeria obtusa* (Schwein) and *Botryosphaeria rhodina* (Berk. and M.A. Curtis), which have been reported in Mexico and the USA [7]; *Diplodia mutila* in the USA [9]; *Dothiorella iberica* in the USA [8], *Lasiodiplodia theobromae* in Spain [10] and Tanzania [11]; *Neofusicoccum australe* in Spain [10] and the USA [12]; *Neofusicoccum luteum* in Spain [10] and the USA [12]; *Neofusicoccum mediterraneum* in Spain [10]; *Neofusicoccum nonquaesitum* in the USA [13]; *Neofusicoccum parvum* in Greece [14], Italy [15], and Spain [16]; and *Neofusicoccum stellenboschiana* in Greece [14].

In Chile, these diseases have been associated with N. australe [17], Diplodia mutila, Diplodia pseudoseriata, Diplodia seriata, Dothiorella iberica, Neofusicoccum nonquaesitum, and Neofusicoccum parvum [5]. However, the predisposing factors for branch canker and dieback have not yet been reported in Chilean orchards.

Avocados with branch cankers develop cankers on the trunk, branches, and twigs, causing friable bark, often with whitish to brownish exudates of perseitol, a crystalline polyhydric alcohol produced by some plant species, such as avocados [6,7]. Dieback occurs in small twigs, and the twigs retain dead leaves that turn brown and fruits that turn completely black with advanced stages of softening and may remain for several months on these twigs [8]. Infection mainly occurs on injured wood, so it has been suggested that wounds associated with pruning, girdling, chilling injuries, mechanical damage, wind, or grafting could allow pathogens to enter [18]. If the infection reaches the vascular tissue of a tree, it can block water and nutrient transport in the xylem and translocation of assimilate reserves to sinks. This flow interruption causes the weakening and decay of the wood at the infection site, which can eventually lead to wilting or death of the branch [19]. Additionally, avocado trees accumulate reserves in their bark; therefore, disruptions in assimilate flow can affect the accumulation and availability of reserves located on the trunk and branches of a tree that are used in the following season for fruiting [20].

Botryosphaeriaceae species are fungi associated with the endophytic microbiome of avocado trees [21]. These species have been considered opportunist pathogens because no damage has been observed in healthy trees, but the post-latent phase can rapidly cause disease when the host is under stress, which increases tree susceptibility [22,23]. The stressors associated with these diseases are drought, extreme temperatures, nutritional deficiencies, flooding, and biotic stresses associated with insects or other pathogens [6–8,19,23].

The wood disease cycle caused by Botryosphaeriaceae species is influenced by the perennial nature of the crops and by the climate. In the Mediterranean climate, Botryosphaeriaceae species can overwinter and oversummer as pycnidia and pseudothecia on dead bark, twigs, and branches [24]. Several studies on trapped airborne spores have indicated that conidia are disseminated mainly by rain splashing [19,25–27], which can cause primary infections during early spring and summer or secondary infections during late summer and autumn [28].

Avocados are subtropical species, and their cultivation under Chilean Mediterranean climate conditions requires the use of crop management techniques to adapt to local water, soil, and climate conditions. For example, avocado production in Chile depends on irrigation, frost protection, and soil irrigation management to avoid root asphyxiation. Additionally, in some areas, the application of the leached water fraction is necessary to avoid salinity problems [29,30]. Moreover, a tendency toward rainfall reduction in the avocado production area of Chile has been observed in the last thirteen years, which has led to diminished water availability for irrigation and thus increased stress conditions for avocado orchards. This fact has been concomitant with increased cases of Botryosphaeriaceae infection in avocados, which has led farmers to think that the increased number of infections is a consequence of drought. Today, there is an increasing demand for avocados

in local and international markets, causing the production areas to move to southern areas of Chile [2], with more water availability but different climate and soil conditions. Thus, it is necessary to understand the predisposing factors of these diseases and to take actions to prevent or mitigate the disease cycle of these Botryosphaeriaceae species.

Considering the opportunistic nature of this group of pathogens, we hypothesize that branch canker and dieback incidence are highly dependent on a combination of stress-inducing orchard management practices and edaphoclimatic conditions that affect the physiological performance of the trees. Therefore, the objective of this study was to determine which climatic conditions, planting features, and/or orchard management variables are associated with the incidence of branch canker and dieback in Chilean avocado orchards.

## 2. Materials and Methods

This study was conducted in sixteen "Hass" avocado orchards selected in the main Chilean avocado production regions. Three different agroclimatic zones were encompassed based on the classification of Pérez and Adonis (2012) [31]. All sites corresponded to irrigated orchards and were located in an area of 428.97 km between Illapel (31°37′ S) and Peumo (34°24′ S) (Figure S1). In the pre-mountain valley, five orchards were selected (Illapel, Alicahue 1, Alicahue 2, San Felipe 1, and San Felipe 2); in the inner valley, four orchards were selected (María Pinto, Nogales 1, Nogales 2, and Ocoa); and in the inner valley with coastal influence, seven orchards were selected (Jaururo, La Ligua, Melipilla 1, Melipilla 2, Peumo, Quillota 1, and Quillota 2). For each agroclimatic zone, orchards with symptoms of wood diseases and orchards without symptoms (apparently healthy orchards) were included. The orchards with wood diseases had avocado trees with symptoms, such as branches with wilted leaves and inflorescences and small black dried fruits, as well as cankers and inner tissue damage in branches and trunks (Figure 1). These symptoms were previously reported by Valencia et al. (2019) [5] as being caused mainly by species of Botryosphaeriaceae in Chilean avocado orchards. A brief description of geographic references and the health conditions of the orchards in each agroclimatic zone is provided in Table 1.

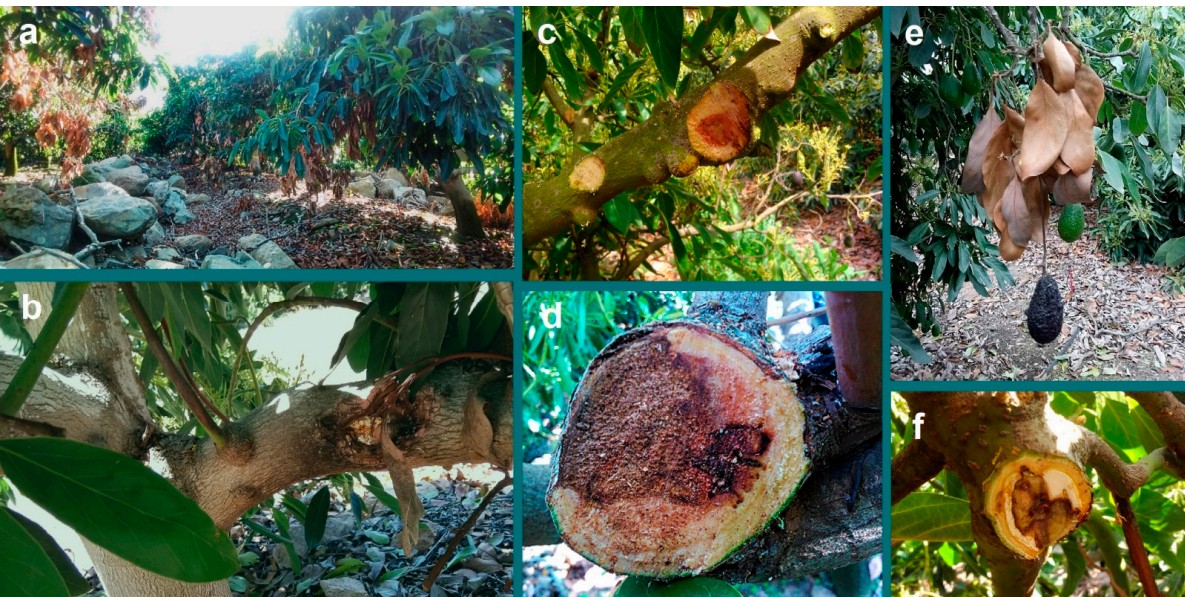

**Figure 1.** Symptoms of branch canker and dieback. (**a**) Chilean orchard with high levels of branch canker and dieback incidence. (**b**) Cankers and friable bark on a main branch near the trunk. (**c**) Branch with protuberances and internal necrotic tissue. (**d**) Internal necrosis in the inner tissue of a trunk associated with canker. (**e**) Branch with dieback with wilted leaves and small black dried fruits. (**f**) Inner tissue damage due to dieback.

**Table 1.** A summary description of the health conditions, agroclimatic zones, and geographic locations of the 16 studied orchards based on the classification of Pérez and Adonis (2012).

| No. | Orchard | Health Condition | Agroclimatic Zone | Latitude | Longitude | Altitude (m) |
|---|---|---|---|---|---|---|
| 1 | San Felipe 1 | Symptomatic | Pre-mountain valley | 32.7336 S | 70.833 O | 706 |
| 2 | Nogales 1 | Symptomatic | Inner valley | 32.7678 S | 71.157 O | 265 |
| 3 | Nogales 2 | Healthy | Inner valley | 32.7708 S | 71.155 O | 264 |
| 4 | Quillota 1 | Healthy | Inner valley with coastal influence | 32.8921 S | 71.1871 O | 211 |
| 5 | San Felipe 2 | Healthy | Pre-mountain valley | 32.7361 S | 70.8446 O | 741 |
| 6 | Ocoa | Symptomatic | Inner valley | 32.8362 S | 71.0442 O | 342 |
| 7 | Jaururo | Healthy | Inner valley with coastal influence | 32.4680 S | 71.3126 O | 131 |
| 8 | María Pinto | Healthy | Inner valley | 33.4545 S | 71.2013 O | 358 |
| 9 | Alicahue 1 | Symptomatic | Pre-mountain valley | 32.3967 S | 70.8640 O | 501 |
| 10 | Alicahue 2 | Healthy | Pre-mountain valley | 32.4022 S | 70.8592 O | 590 |
| 11 | Melipilla 1 | Healthy | Inner valley with coastal influence | 33.7694 S | 71.1928 O | 185 |
| 12 | Illapel | Symptomatic | Pre-mountain valley | 31.596 S | 71.0793 O | 524 |
| 13 | La Ligua | Symptomatic | Inner valley with coastal influence | 32.4557 S | 71.2124 O | 71 |
| 14 | Quillota 2 | Symptomatic | Inner valley with coastal influence | 32.8856 S | 71.2034 O | 148 |
| 15 | Peumo | Symptomatic | Inner valley with coastal influence | 34.3853 S | 71.2111 O | 161 |
| 16 | Melipilla 2 | Symptomatic | Inner valley with coastal influence | 33.7652 S | 71.1228 O | 226 |

This inferential study was developed during two consecutive growing seasons (2014 and 2015), from September 2014 to August 2015 and from September 2015 to August 2016. This period is associated with the phenological stage between avocado tree flowering and harvest.

During each growing season (2014 and 2015), commercial Hass avocado orchards were characterized, collecting information on 76 variables, corresponding to the main characteristics of each orchard, historic reports of biotic and abiotic stresses, and specific conditions or cultural management of orchards during the seasons in the study (for details, please see Table S1). We include data from the two previous consecutive seasons to expand the dataset of each variable, in order of considering the recent history of the orchard prior to the sampling and identification of CD.

*2.1. Study Variables*

2.1.1. Climate Variables

The climate data were obtained from meteorological stations near the avocado orchards selected in this study. The data collection included data recorded every hour, and with these data, seasonal average data were calculated. The seasons were considered as follows: spring (sp, September to December), summer (su, December to March), autumn (au, March to June), and winter (wi, June to September). The variables analyzed in each season were average air temperature (AT, °C), maximum air temperature (MAXT, °C), minimum air temperature (MINT, °C), average solar radiation (RAD, Wm$^{-2}$), average relative air humidity (Rh, %), and accumulated precipitation (Pp, mm).

2.1.2. Planting Features

The variables collected related to orchard edaphic and planting conditions were: location (latitude (Lat), longitude (Long), altitude (Alt), agroclimatic zone (Zone)), row orientation, plantation density (Plants/ha), age of each plant (PlantAge), yield, volume of the canopy (VolumeC), diameter of the trunk (DiameterT), rootstock variety (Rstock), leaf area index (LAI), soil characteristics (texture, bulk density (Bd), pH (pHS), electrical conductivity (ECS), organic matter (OMS), nitrogen (NS), phosphorus (PS), potassium (KS), cationic exchange capacity (CECS)), foliar nutrient content (nitrogen (NF), phosphorus (PF), potassium (KF), calcium (CaF), magnesium (MgF), copper (CuF), manganese (MnF),

and zinc (ZnF)), crop evapotranspiration in each year season (ETcsp, ETcsu, ETcau, and ETcwi; spring, summer, autumn, and winter, respectively), and the background biotic (pest and diseases) and/or abiotic stressors (i.e., frost). To analyze the levels of water deficit or overirrigation in each orchard, the percentage of water applied with respect to the real requirement (evapotranspiration, ETc) in summer (H2OETcsu) was calculated.

### 2.1.3. Crop Management Variables

The management variables considered in this study were: the irrigation system (Isystem, drip or micro-sprinkler), pruning management (date (DateP), frequency of pruning (FrP), intensity of pruning (IP), seal used for pruning wounds (PasteP), application of fungicide (FungicideP)), nutrition (applied dosage of nitrogen (UN/ha) and applied dosage of calcium (Ca+)), application of humic acids (HumicA) and application of plant growth regulators (annual doses applied of growth regulator (GR), site of growth regulator application (GRsite, foliage or soil), and date of growth regulator application (Grdate)). Girdling management is a common practice used in Chilean avocado orchards; however, in this study, only one orchard used this technique. Therefore, we did not include girdling as a variable in this study.

### 2.1.4. Disease Index

The incidence of branch canker and dieback (Icd) for each orchard was determined by the percentage of trees with previously indicated symptoms of wood diseases with respect to the total number of trees existing in the study area. Also included in this analysis was the distribution of symptomatic trees in this area (uniform, random, and aggregate). The severity of CD was determined using the severity scale, which included: 0 = healthy trees; 1 = low dieback, without branch canker; 3 = low dieback, low branch canker; 5 = low dieback, moderate branch canker; 7 = high dieback, low branch canker, and 9 = low dieback, high branch canker).

The study area was of approximately 900 m$^2$, a surface area representative of the planting zones in the orchards. This surface was the baseline of the sampling area because it included at least 15 trees; the orchards considered in this study comprised different planting distances (178–1656 plants/ha). To corroborate that the symptoms observed were associated with fungal pathogens of these wood diseases, samples from each orchard in the study were transported to the laboratory. The wood samples were cut, disinfected by immersion in 96% ethanol for 15 s, and flamed. Then, small pieces of wood were deposited in Petri dishes with 2% potato dextrose agar acidified with 0.5 mL Liter$^{-1}$ of 92% lactic acid plus 0.05% tetracycline. The cultures were incubated at 20 °C in darkness for 7 and 21 days. Pure cultures were obtained from hyphal tip transfers. The colonies, conidiophores, and conidia were used to identify the morphological isolates obtained. In a previous study, the isolates in this study were identified by DNA analysis to determine the species of Botryosphaeriaceae [5].

### 2.2. Statistical Analysis

A dataset of 16 orchards and 76 variables (total data = 2304) included in this study was analyzed under three scenarios. These scenarios considered data in each orchard during the 2014 and/or 2015 seasons. Scenario 1 included data from the 2014 season and the 2015 season, scenario 2 included only data from the 2014 season, and scenario 3 included only data from the 2015 season. The growing seasons were analyzed separately and together to corroborate that the correlation structure by season was replicated in the joint scenario.

Different multivariate analyses were performed to study the dataset, including variables associated with climate, planting features, management variables, and disease incidence. Chemometrics methods were used because the global aim of this type of multivariate analysis is to determine the relationship between variables and observations, called the "correlation structure" (CS) [32]. Moreover, Eriksson et al. (2013) [32] expressed this

approach as follows: "multivariate analysis based on so-called projection methods represents the observations from a phenomenon as a swarm of points in a K-dimensional space (K = number of variables), and then projects the point swarm down onto a lower-dimensional hyperplane". The projection approach can be adapted to a variety of analytical objectives, i.e., (i) summarizing and visualizing a dataset, (ii) multivariate classification and discriminant analysis, and (iii) finding quantitative relationships among variables. This approach applies to any shape of multivariate data, with many or few variables, many or few observations, and complete or incomplete data tables (missing data). In particular, the projections handle data matrices with more variables than observations very well, and the data can be noisy and highly collinear.

The methods applied were principal component analysis (PCA) and partial least squares discriminant analysis (PLS-DA). These analyses were based on the nonlinear iterative partial least squares algorithm (NIPALS) [33], which allows the analysis of a large number of highly correlated variables and ill-conditioned data, that is, an incomplete rank matrix (dataset with more columns than rows) [34]. All variables were centered and standardized to the unit variance prior to the analysis. All models were validated by a full cross-validation routine, minimizing the prediction residual sum of squares statistics (PRESS) to avoid overfitting the model [35].

Additionally, an overview variance plot was used to evaluate the different results. Thus, it was possible to identify the variables that best contributed to the distribution and conformation of groups (clusters) from the scores in the projection hyperplane.

Before multivariate analysis was conducted, the data of some variables included in this study, such as Zone, Rstock, texture, Isystem, DateP, FrP, IP, Grsite, Grdate, Pest, Diseases, Distribution, and SCD, were transformed into ordinal scales (1, 3, 5, 7 . . . n) according to the procedures described by Knapp (1990) [36] and Norman (2010) [37] (Table S1). In addition, other variables, such as PasteP, FungicideP, Ca+, HumicA, and Frost, were analyzed as quality variables (presence "1" or absence "0") [38]. Both treatments were applied since multivariate algorithm-based (NIPALS) analysis is a free-probabilistic distribution method [39,40].

PCA was performed to determine the relationships between the climate, planting, and management variables and disease incidence. PCA is a method that allows synthesizing of information contained in a large matrix of variables within a smaller set of factors (principal components) with minimal loss of information [41]. Likewise, PCA allows for condensing the information in two ways: identifying relationships between the observations (orchards in study) that comprise the score matrix and determining relationships between variables, known as the loadings. Moreover, PCA allows us to display the relationships between observations and variables in orthogonal planes that represent the direction of the greatest variance contained in the dataset [42,43].

According to the $R^2VX$ value (fraction of independent variance explained for every variable of X), a group of classes was tested in the PLS-DA. Thus, it was possible to predefine three classes to classify the orchards. For this purpose, the variables with $R^2VX$ values greater than 0.5 were discriminated and used in this analysis [34].

The PLS-DA allowed us to know the relationships among the orchards grouped in the classes and all variables to determine the causality of possible significant groups associated with problem variables that are responsible for the behavior of that class and whether there are anomalous or, eventually, new classes [44–46].

All chemometric methods were performed using SIMCA-P v.14 software (Umetrics AB, Malmo, Sweden).

## 3. Results

### 3.1. Disease Incidence

Of the 16 studied sites, nine orchards showed symptoms associated with branch canker and dieback (CD). In this study, in all the symptomatic orchards, the branch canker began with rough protuberances on the bark of the trunk and twigs of a tree associated with the

internal necrotic tissue (Figure 1). In addition, in symptomatic orchards, different incidence levels of CD were observed. The orchards that had high levels of Icd (71 to 100%) were San Felipe 1 (71%), Quillota 2 (72%), Illapel (100%), Melipilla 2 (100%), and Peumo (100%). In addition, the orchards with low levels of Icd (31 to 59%) were La Ligua (31%), Nogales 1 (37%), Ocoa (39%), and Alicahue (59%). The other seven orchards without CD symptoms (Alicahue 2, Jaururo, Maria Pinto, Melipilla 1, Nogales 2, Quillota 1, and San Felipe 2) were considered healthy (Table 2, Figure S1).

**Table 2.** Incidence of branch canker and dieback and the species of Botryosphaeriaceae associated with each Chilean orchard in this study.

| | Orchard | | | | | | | | | | | | | | | |
|---|---|---|---|---|---|---|---|---|---|---|---|---|---|---|---|---|
| | 1 | 2 | 3 | 4 | 5 | 6 | 7 | 8 | 9 | 10 | 11 | 12 | 13 | 14 | 15 | 16 |
| **Incidence (%)** | 71 | 37 | 0 | 0 | 0 | 39 | 0 | 0 | 59 | 0 | 0 | 100 | 31 | 72 | 100 | 100 |
| **Botryosphaeriaceae** | Isolates (n = 36) | | | | | | | | | | | | | | | |
| *Diplodia* sp. | 0 | 0 | 0 | 0 | 0 | 0 | 0 | 0 | 1 | 0 | 0 | 0 | 0 | 0 | 0 | 0 |
| *D. mutila* | 0 | 0 | 0 | 0 | 0 | 3 | 0 | 0 | 0 | 0 | 0 | 0 | 0 | 0 | 0 | 0 |
| *D. pseudoseriata* | 0 | 0 | 0 | 0 | 0 | 0 | 0 | 0 | 0 | 0 | 0 | 0 | 0 | 1 | 0 | 0 |
| *D. seriata* | 1 | 0 | 0 | 0 | 0 | 0 | 4 | 0 | 0 | 0 | 0 | 0 | 0 | 0 | 0 | 0 |
| *Dothiorella iberica* | 2 | 0 | 0 | 0 | 0 | 0 | 0 | 0 | 5 | 0 | 0 | 0 | 0 | 0 | 1 | 0 |
| *Neofusicoccum* sp. | 0 | 1 | 0 | 0 | 0 | 0 | 0 | 0 | 0 | 0 | 0 | 0 | 0 | 0 | 0 | 0 |
| *N. australe* | 0 | 0 | 0 | 0 | 0 | 1 | 0 | 0 | 0 | 0 | 0 | 0 | 1 | 0 | 0 | 0 |
| *N. nonquaesitum* | 0 | 0 | 0 | 0 | 0 | 2 | 0 | 0 | 0 | 0 | 0 | 0 | 0 | 1 | 0 | 0 |
| *N. parvum* | 0 | 0 | 0 | 0 | 0 | 8 | 0 | 0 | 0 | 0 | 0 | 0 | 0 | 0 | 4 | 0 |
| **Others** | Isolates (n = 10) | | | | | | | | | | | | | | | |
| *Pestalotiopsis* sp. | 3 | 2 | 0 | 0 | 0 | 0 | 0 | 0 | 1 | 0 | 1 | 0 | 0 | 1 | 0 | 0 |
| *Colletotrichum* sp. | 0 | 0 | 0 | 0 | 0 | 0 | 1 | 0 | 0 | 0 | 0 | 0 | 0 | 0 | 0 | 0 |
| *Diaporthe* sp. | 0 | 0 | 0 | 0 | 0 | 0 | 0 | 0 | 0 | 0 | 0 | 0 | 0 | 0 | 1 | 0 |

Most species found belonged to the Botryosphaeriaceae, and the number of species associated with these diseases varied according to the orchard. For example, in the Ocoa orchard, three different species were obtained (*D. mutila*, *N. australe*, and *N. nonquaesitum*), while in other orchards, either one or two Botryosphaeriaceae species were isolated from wood. In contrast, from the Melipilla 2 and Illapel orchards, which are orchards with low and high levels of Icd, respectively, it was not possible to isolate Botryosphaeriaceae species or other species associated with wood diseases. From the orchard located in Jaururo, which did not show symptoms of CD, four isolates of *D. seriata* were obtained from the wood samples (Table 2).

*Dothiorella iberica* was the species most frequently found in this study and was obtained from the wood samples of Quillota 1, Alicahue 1, and Peumo. In some cases, the same species was found in two orchards: *D. seriata* was found in San Felipe 1 and Jaururo, *N. australe* was found in Ocoa and La Ligua, *N. nonquaesitum* was found in Ocoa and Quillota, and *N. parvum* was found in Ocoa and Peumo. Additionally, there were species found only in one orchard: *D. pseudoseriata* in Quillota 2, *D. mutila* in Ocoa, *Diplodia* sp. in Alicahue 1, and *Neofusicoccum* sp. in Nogales (Table 2).

Additionally, ten isolates of other fungal pathogen species were detected, all associated with avocado diseases. Isolates of *Pestalotiopsis* sp. were isolated from wood samples of San Felipe 1, Nogales 1, Alicahue 1, Melpilla 1, and Quillota 2. Additionally, one isolate of *Colletotrichum* sp. was isolated from a wood sample of Jaururo, and one isolate of *Diaporthe* sp. was obtained from Peumo.

### 3.2. Multivariate Analysis

The initial PCA incorporated the dataset of 16 orchards and 76 variables of data obtained during the 2014 season and the 2015 season (Figure 2). This multivariate analysis allowed us to select variables that can explain the model because the variables with

$R^2VX \leq 0.5$, namely, variables with low variance, must be set aside from the datasets to maximize the explicative capability of the models.

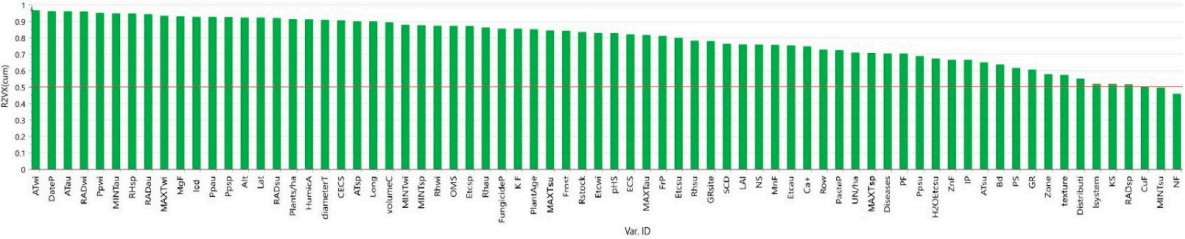

**Figure 2.** Weight plot of the PCA variables associated with the incidence of branch canker and dieback in Chilean orchards prospected in this study in the 2014 and 2015 seasons, including 16 orchards and 76 variables. $R^2VX$: fraction of independent variance explained for every variable X. Red line shows the limit criteria for explained variance ($R^2VX \leq 0.5$).

PCA of scenario 1. The PCA performed included data analysis of both growing seasons (2014 and 2015). This analysis indicated that the multifactorial model retained two components that explained 77.9% of the total variance. Factor 1 explained 62.4% of the total variance, while Factor 2 explained 15.6% of the total variance. In the score and loading plots, Factor 1 sorted the orchards on the left side according to their high values of VolumeC, DiameterT, PlantAge, Icd, and Frost; on the right side, Factor 1 indicated that the variables Plant/ha and LAI had inverse relationships with Icd. Factor 2 sorted the orchards according to H2OETcsu (Figure 3).

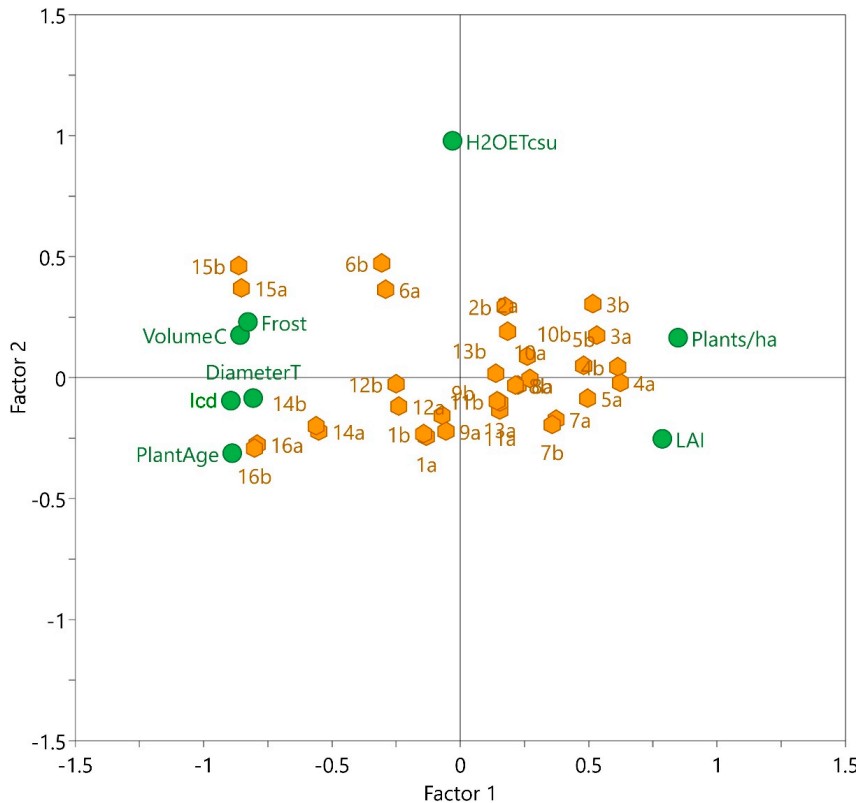

**Figure 3.** Biplot of the principal component analysis of predisposing factors associated with the incidence of branch canker and dieback in Chilean orchards prospected in this study in the 2014 and 2015 seasons. Scores (orange hexagons) are shown for the sixteen orchards in both growth seasons (32 observations; 2014 season (a); 2015 season (b)). Loadings are shown in green circles. $R^2X$ Factor 1 = 62.4%; $R^2X$ Factor 2 = 15.6%; $R^2Xcum$ = 77.9%.

PLS-DA analysis of scenario 1. This analysis extracted three components that explained 75.4% of the total variance in matrix Y ($R^2Y$), and the cumulative overall cross-validated $Q^2$cum (the cumulative fraction of the total variation that can be predicted by specific factors) was 64.5% for the three previously defined classes. According to the distribution of variables and observations in the biplot graph of PLS-DA, three classes were previously defined: Class 1 included orchards with high levels of CD incidence (71 to 100%) and high VolumeC; Class 2 included orchards with low levels of CD incidence (31 to 59%) and high Frost, PlantAge, and H2OETcsu values; and Class 3 included orchards without CD incidence and with a high LAI and Plants/ha (Figure 4).

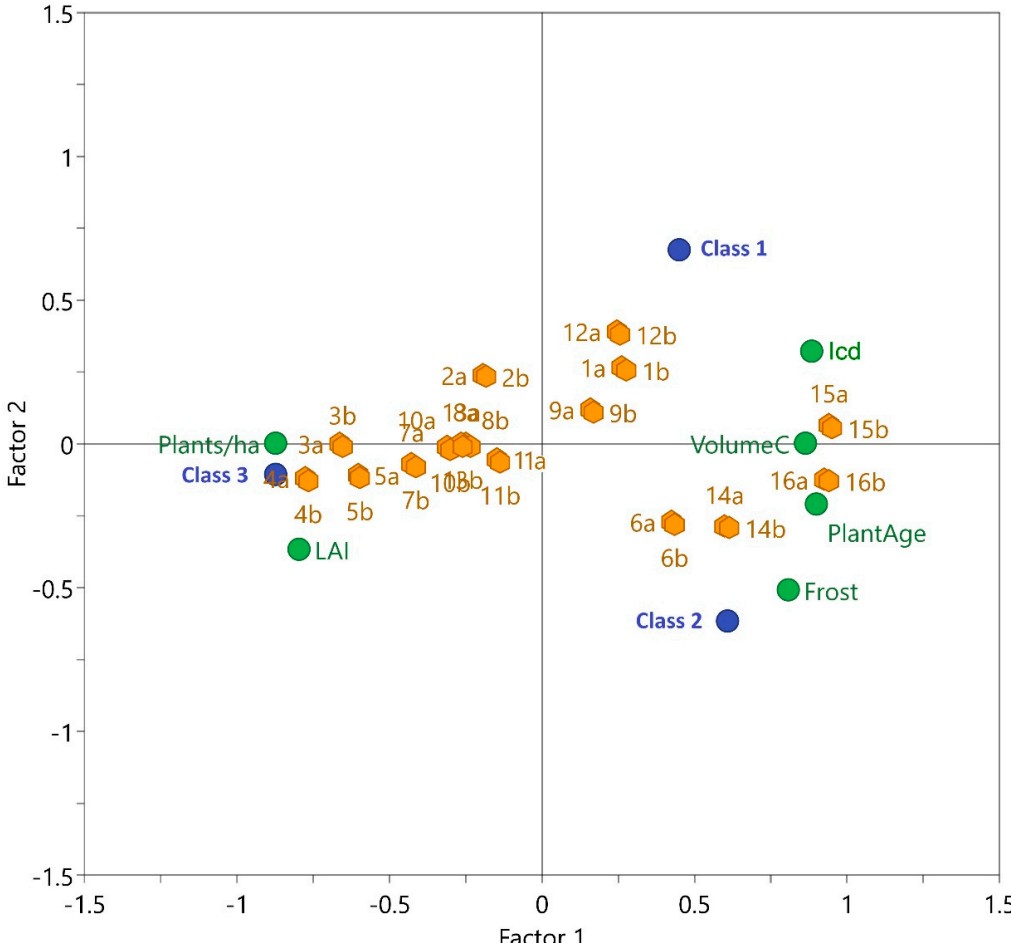

**Figure 4.** Biplot of the partial least squares discriminant analysis of predisposing factors associated with the incidence of branch canker and dieback in the sixteen orchards prospected in this study in the 2014 and 2015 seasons. Scores (orange hexagons) are shown for the sixteen orchards in both growth seasons (32 observations; 2014 season (a); 2015 season (b)). Loadings are shown in green circles. Factors = 3; $R^2$Ycum = 75.4%; $Q^2$Ycum = 64.5%. Class = 3; Class 1: Icd, VolumeC; Class 2: PlantAge, Frost, H2OETcsu; Class 3: Plants/ha, LAI.

PCA of scenario 2. The PCA performed under scenario 2 included data analysis of the 2014 growth season. This analysis indicated that the multifactorial model retained four components that explained 86.3% of the total variance. Factor 1 explained 45.5% of the total variance, while Factor 2 explained 24.8% of the total variance. In the score and loading plots, Factor 1 sorted the orchards according to their geographic zones. On the right side, the orchards were sorted with high values for the variables Long, Rhsp, Rhsu, Rhau, and Rhwi, which were associated with the orchards localized in the inner valley with a coastal influence, and some orchards were sorted on the left side with high values for the

variables Alt, RADsu, RADsp, MINTsp, ATsu, ATsp, and KF, which were associated with the orchards located in the pre-mountain valley. On the top side, Factor 2 sorted orchards with low levels of Icd and high values for the variables RADwi and MINTwi; on the bottom side, Factor 2 sorted the orchards with high levels of Icd and high Lat, VolumeC, Ppsp, H2OETcsu, and Frost (Figure S2).

PLS-DA analysis of scenario 2. This analysis extracted two components that explained 82.2% of the total $R^2Y$, and the cumulative overall cross-validated $Q^2$cum was 69.4% for the three previously defined classes: Class 1 included orchards with and without CD and high values for Long, Rhsp, Rhsu, Rhau, Rhwi, RADwi, and MINTwi; Class 2 included orchards with and without CD and high values for Alt, MINTsp, RADsp, and KF; and Class 3 included orchards with high levels of CD incidence and high values for the variables Lat, VolumeC, Frost, Ppsp, and H2OETcsu (Figure 5).

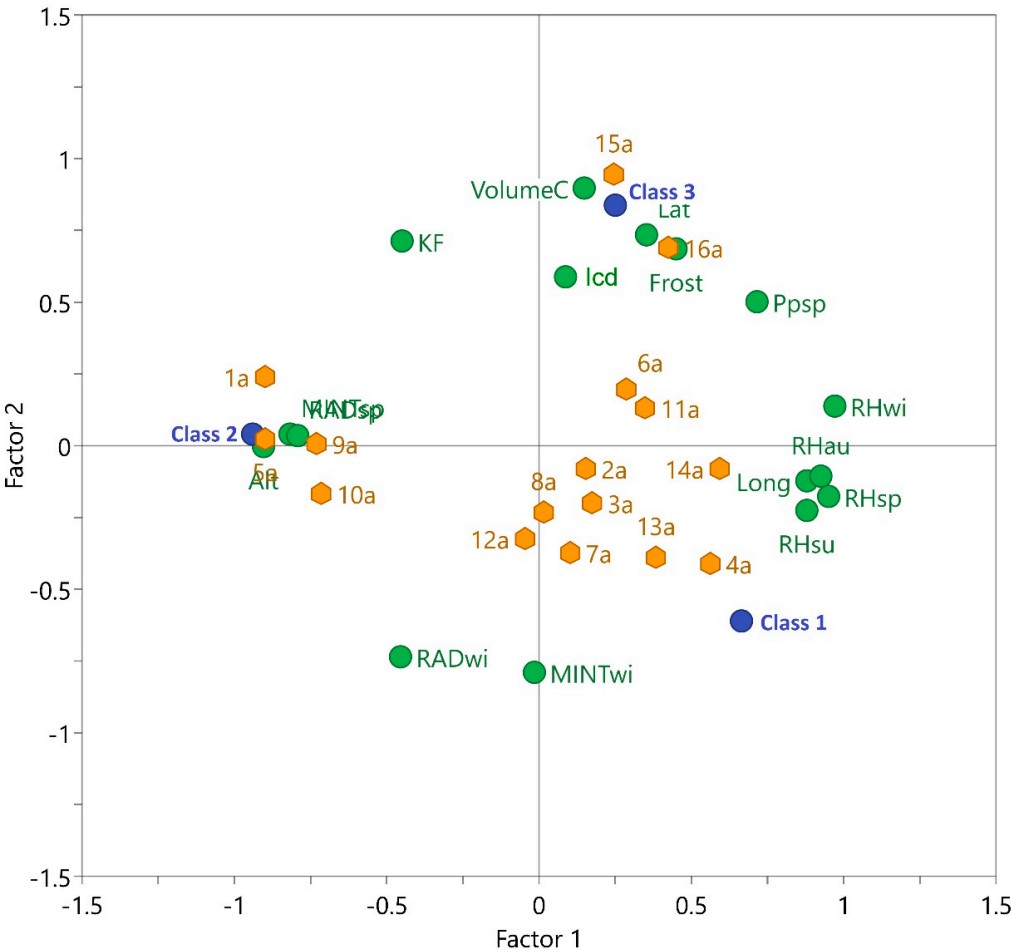

**Figure 5.** Biplot of the partial least squares discriminant analysis of predisposing factors associated with the incidence of branch canker and dieback in the sixteen orchards prospected in this study in the 2014 season. Scores (orange hexagons) are shown for the sixteen orchards (16 observations (a)). Loadings are shown in green circles. Factors = 2; $R^2Y$cum = 82.2%; $Q^2Y$cum = 69.4%. Class = 3; Class 1: Long, Rhsp, Rhsu, Rhau, Rhwi, RADwi, and MINTwi; Class 2: Alt, MINTsp, RADsp, and KF; Class 3: Lat, volumeC, Icd, Frost, Ppsp, and H2OETcsu.

PCA of scenario 3. The PCA performed under scenario 3 included data analysis of the 2015 growing season. This analysis indicated that the multifactorial model retained three components that explained 85.2% of the total variance. Factor 1 explained 45.7% of the total variance, while Factor 2 explained 31.7% of the total variance. In the score and loading plots, Factor 1 sorted the orchards according to their geographic zones. On the right side, it

sorted the orchards with high values for Long, MINTau, and Ppwi, which were associated with the orchards localized in the inner valley with a coastal influence. On the left side, this factor sorted the variables Alt, MAXTau, MINTsu, and ATsu, which were associated with the orchards located in the pre-mountain valley. Factor 2 sorted the orchards according to their incidence levels of disease; on the top side, this factor sorted the orchards with high levels of Icd and high values for the variables Rstock, PlantAge, DiameterT, and volumeC. Factor 2, on the bottom side, sorted the orchards with low levels of disease and high values for the variable Plants/ha (Figure S3).

PLS-DA analysis of scenario 3. This analysis extracted three components that explained 85.4% of the total $R^2Y$, and the cumulative overall cross-validated $Q^2$cum was 72.0% for the three classes previously defined. Class 1 included orchards with high levels of CD incidence and high values for DiameterT, VolumeC, PlantAge, and Rstock; Class 2 included orchards with and without CD and high values for ATsu, Alt, MINTsu, and MAXTau; and Class 3 included orchards with low levels and without CD incidence and high values for Plants/ha, Long, Ppwi, Ppsp, and MINTau (Figure 6).

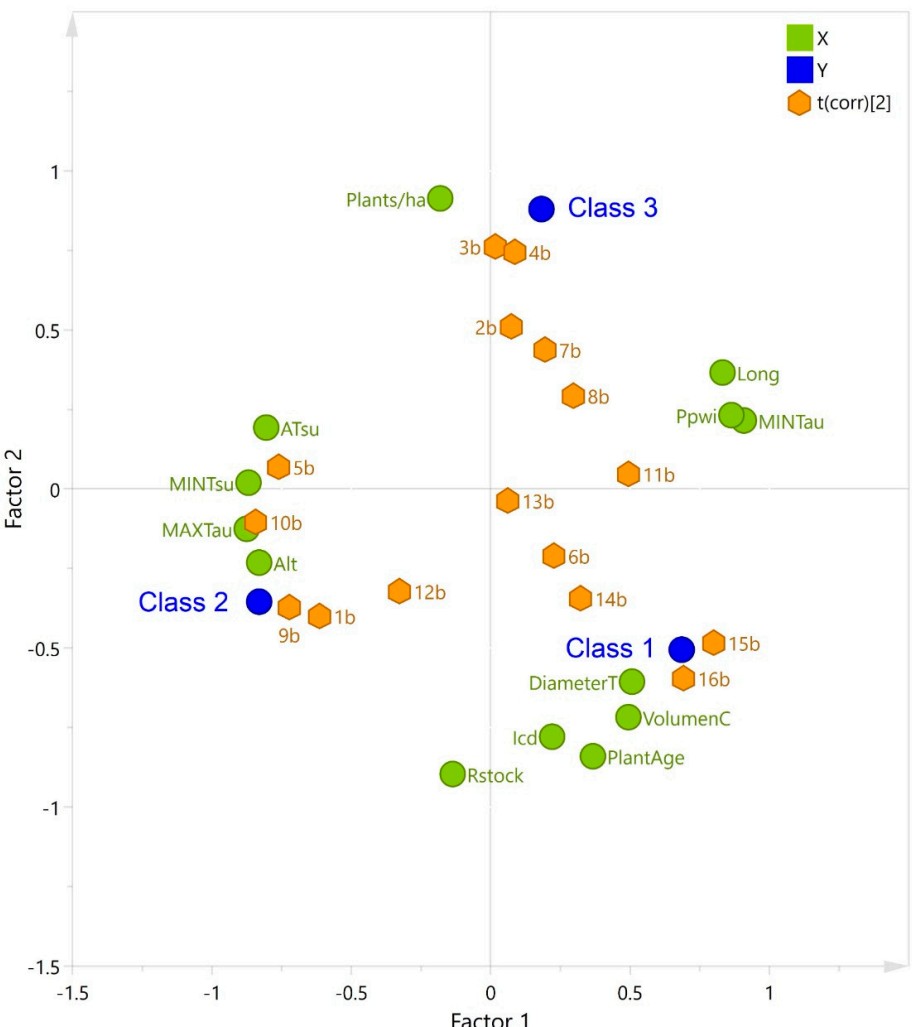

**Figure 6.** Biplot of the partial least squares discriminant analysis of predisposing factors associated with the incidence of branch canker and dieback in the sixteen orchards prospected in this study in the 2015 season. Scores (orange hexagons) are shown for the sixteen orchards (16 observations (b)). Loadings are shown in green circles. Factors = 3; $R^2Y$cum = 85.4%; $Q^2Y$cum = 72.0%. Class = 3; Class 1: DiameterT, VolumeC, PlantAge, and Rstock; Class 2: ATsu, Alt, MINTsu, and MAXTau; Class 3 Plants/ha, Long, Ppwi, Ppsp, and MINTau.

## 4. Discussion

This case study is the first multivariate analysis of the climate, planting, and crop management variables associated with the incidence of branch canker and dieback in an avocado-producing country in the world. To date, the main factors associated with these diseases remain unclear; however, the disease incidence levels obtained from Chilean orchards indicated that, in the same agroclimatic zone, it is possible that these diseases can or cannot be developed because there were high, low, and zero levels of Icd in the pre-mountain valley, inner valley, and inner valley with a coastal influence, respectively. Thus, the incidence of wood diseases could be mainly associated with the planting features, climatic conditions, and management of each orchard together with the presence of latent pathogen species. Pathogenic members of Botryosphaeriaceae were the most frequent species associated with CD detected in Chilean orchards in this study, with 36 isolates corresponding to seven species reported as pathogens causing wood disease in Chilean avocado orchards [5]. Moreover, eight isolates of *Pestalotiopsis* sp. were detected from wood samples in some of the studied orchards; however, in Chile, this species has been reported only as a pathogen causing stem end rot in avocado fruit [47]. In addition, one isolate of *Colletotrichum* sp. and one isolate of *Diaporthe* sp. were obtained from some of the wood samples; these species have been considered by McDonald and Eskalen (2011) [8] as sporadically isolated fungi from avocado branch cankers.

Symptoms associated with orchards affected with the wood disease were branches with wilted leaves and inflorescences and small black dried fruits, as well as cankers and inner tissue damage in branches and trunks. We also observed rough protuberances on the avocado bark of the trunk and twigs with CD; however, these symptoms have not yet been reported as symptoms of branch canker in avocado trees. In the cases of blueberry plants, it has been reported that protuberances in the bark are the consequence of new periderm tissue produced in response to infections by Botryosphaeriaceae species [48]. During this work, we did not study these symptoms more deeply, but it would be interesting to address them in the future to shed light on the importance of a new diagnostic tool.

In this study, PCA was performed as an exploratory tool that allowed us to visualize the underlying variations among the orchards and variables and possible clusters. Later, considering the contribution of variables in each orchard, PLS-DA was performed as a multivariate linear regression method, after which the orchards were grouped and thus identified as belonging to three different classes. In this study, six to eight selected variables (those with the higher explicative capability) were identified with PLS-DA, which is consistent with mainstream literature results such as those from Brereton (2009) [49] and Eriksson et al. (2013) [32] that justified that different phenomena can be explained by a small group of variables, commonly called underlying-projection-data-based, in the form of a few new variables (linear combination of the original variables), rather than by the total original variables based on the parsimony principle. Some agricultural studies on avocados have included this multivariate analysis to identify factors associated with postharvest conditions [50,51].

According to the results, the PCA performed in scenario 1 (seasons 2014 and 2015) showed that orchards with high levels of Icd corresponded to orchards with aged or stressed trees, characterized by older tree ages, higher canopy volumes but lower leaf area indices, large trunk diameters, and orchards with high frost risks. In contrast, orchards with low or non-CD incidence were associated with high leaf area indices and high plant densities. For the same scenario, the PLS-DA indicated that, according to the contributions of the variables and scores, there were three classes, grouped as Class 1, Class 2, and Class 3, that were coincident in this scenario with high levels of CD incidence, low levels of CD incidence, and the absence of CD, respectively. As seen after PCA, the orchards in classes with CD incidence had large canopy volumes, high frost, and high plant ages (Class 2). The group indicating no CD incidence (Class 3) included orchards with high leaf area indices and high planting densities, which indicated vigorous or new orchards. The multivariate PCA of scenario 2 (2014 season) and scenario 3 (2015 season) confirmed the

results obtained after analyzing scenario 1: the CD incidence was highest in orchards with aged or stressed trees. In Scenario 2, more information was obtained after PLS-DA since classes grouped as having a high Icd were associated with other variables, such as high latitudes, high frost risks, high spring precipitation levels, and high-water amounts applied with respect to the summer water requirements. In this sense, it is important to indicate that the water amounts applied with respect to the summer water requirements are a secondary variable that allows us to explain the model. The effect of overirrigation on Icd could be attributable as a variable that affects the physiological performance of the trees or because higher water availability was related to orchards being older and more infected trees being found. It is important to indicate that micro-irrigation (drip and micro-sprinkler systems) is highly used as an irrigation system in Chilean orchards; however, overirrigation in avocado orchards is very common and could explain the high levels of CD incidence. Moreover, Eskalen et al. (2013) [19] indicated that the dispersal of Botryosphaeriaceae spores requires the impact of water droplets artificially through micro-sprinkler irrigation, and several spore trapping studies have indicated that infection by micro-sprinkler irrigation causing primary and secondary infections throughout the growth season [24]. This information shows that age, climate conditions, location, and management practices, such as irrigation, can be important factors during growing seasons for CD incidence.

Another interesting result obtained after the PLS-DA of scenario 2 and scenario 3 is that some of the classes sorted the orchards with low and high Icds according to their agroclimatic zones, attributed to their latitudes, longitudes, and altitudes. In the 2014 season, all orchards with low or high CD incidence were localized in the pre-mountain valley and inner valley. In the 2015 season, the orchards localized in the pre-mountain valley, inner valley, and inner valley with a coastal influence also had CD incidence (low or high CD incidence). Hence, the results obtained in scenarios 2 and 3 indicated that CD incidence exists independently of agroclimatic zones. Therefore, CD incidence is not conditioned by the agroclimatic zones of orchards with trees infected by pathogens such as Botryosphaeriaceae species, which coincides with the results of McDonald and Eskalen (2011) [8], who recovered Botryosphaeriaceae species from avocado trees with branch canker from orchards located in northern and southern counties in California.

Identifying pathogen species associated with these diseases is very important because, in this family, some species are more virulent than others, which could be associated with severe damage detected in orchards with high incidence levels. In this sense, *N. parvum*, *N. nonquaesitum*, and *D. pseudoseriata* are considered highly virulent species in pathogenicity testing on avocado plants [5,12,13]. These species were detected in this study in Ocoa, Peumo, and Quillota 2, three orchards more than 18 years old, with severe damage from CD (data not shown) and different levels of CD incidence. Moreover, *Dothiorella iberica* was the species detected most frequently in this study in orchards with low and high levels of CD incidence. In this sense, the difference in the incidence of these diseases in the orchards could be explained by inner favorable conditions for the pathogens in each orchard, which allowed the development of high levels of damage in the trees and greater sources of inoculum and dissemination.

In the case of Melipilla 2, which was an orchard with a high level of CD incidence but without isolates of Botryosphaeriaceae species obtained from the wood samples, it is necessary to indicate that, in this orchard, *N. australe* and *N. parvum* were detected in avocado fruits with stem end rot, as reported in Valencia et al. (2019) [5]. Therefore, the high incidence of CD could be attributed, in this case, to both species; according to Slippers and Wingfield (2007) [23], these species are the most damaging species in the Botryosphaeriaceae family. Additionally, Twizeyimana et al. (2013) [52] demonstrated that isolates of *N. parvum* and *N. australe* obtained from branch canker can be sources of inoculum for fruit infections and lead to the development of stem-end rot. In contrast, from the Illapel orchard (located at an altitude of 524 m in the pre-mountain valley and with no frost risk) with a high level of CD incidence, Botryosphaeriaceae species were not detected in the wood or fruit samples; this situation could be explained by an old infection and

disease development associated with the high level of stress caused by drought occurring in this area (Coquimbo Region), which caused a reduction in planting area of two thousand hectares between 2007 and 2017 [2]. Further, Slippers and Wingfield (2007) [23] indicated that site-specific factors affect the host affinity of endophyte communities and that these communities change spatially and temporally in woody hosts.

The detection of *D. Seriata* in orchards located in Jaururo with zero CD incidence could be associated with the capacity of Botryosphaeriaceae to survive endophytically in trees causing no symptoms for many years [53]. However, Slippers and Wingfield (2007) [23] indicated that these symptoms can develop rapidly and cause extensive losses over large areas when widespread abiotic stress occurs. Therefore, the presence of Botryosphaeriaceae species in the healthy plant tissues of woody hosts has a critical role in the epidemiology of these diseases.

One of our hypotheses was that trees in orchards with higher plantation densities (more than 1000 plants/ha) could be more affected by the studied diseases because of the higher need for pruning; however, the results indicated that higher planting densities, together with higher leaf area indices, were associated with low disease incidence, probably because these factors are associated mainly with orchards that have young and vigorous trees and thus better conditions to block or defend against the development of diseases. This study was performed in orchards planted between 1989 and 2013, with planting densities ranging between 1600 plants per hectare in young orchards and 200 plants per hectare in older orchards. In Chile, high-density planting is currently a common practice that requires plant growth regulator applications (PGRs) and frequent pruning [3]. Our results show that pruning itself is not a determinant of CD incidence and that it is more associated with young orchards. However, it must be considered that in aging plantations, severe pruning could increase the infection risk through pruning wounds and could increase the dissemination of pathogens between trees [8,19]. Therefore, orchards with low levels of CD incidence could develop severe damage due to these diseases over time if the causal agents are present as latent infections.

According to our results, the incidence of avocado wood diseases such as branch canker and dieback was consistently related to aged orchards and to conditions not related to age but probably resulting from management and climate events that can accelerate aging (high canopy volume and thus probably shaded orchards, overirrigation, and frost that caused injuries). It should be noted that this study considered mostly data from variables obtained during the period of study and not conditions that occurred during the previous years; thus, orchards with CD could have been affected by factors that occurred in previous years, such as poorly performed irrigation, anoxia, girdling, or applications of plant growth regulators. Additionally, another factor to take into account in relation to the high CD incidence of aged plants and therefore aged orchards is the accumulation of the pathogens' propagule in the environment over time, because fruiting bodies persist on the tree canopy or the ground [24].

Strategies to reduce these diseases should be aimed at avoiding the premature aging of trees, for which it is necessary to mitigate the effects of environmental stressors, and include annual pruning and enhancing the vigor of the trees through early pruning, maintenance of root health, watering according to the water consumption of the crop, and in the case of orchards with a history of the disease, avoiding the excessive use of plant growth regulators and including disinfection and fungicide applications after pruning and girdling to avoid new infections. Given the high percentage of cases found with the incidence of the studied diseases, it is also necessary to adopt preventive measures, such as disinfection of tools used for pruning and girdling and elimination of the pruning remnants that can serve as inoculum for these diseases. As a curative measure, it is necessary to develop products that make it possible to control the diseases present in orchards, since thus far there are no products with proven efficacy for the treatment of Botryosphaeriaceae in avocado trees.

## 5. Conclusions

The multivariate analysis of wood diseases mainly caused by Botryosphaeriaceae pathogens in consecutive growing seasons allowed us to identify the main factors associated with the incidence of branch canker and dieback in Chilean avocado orchards among a large number of climate, planting, and crop management variables. Our study indicated that these diseases are conditioned mainly by orchard variables such as plant age, canopy volume, trunk diameter, leaf area index, and planting density. Variables associated with aged orchards, such as high age, high canopy volume, and high trunk diameter, appeared to be more associated with high incidence levels than were younger orchards with higher planting densities and vigor levels.

Additionally, this study revealed that these diseases can be developed in different agroclimatic zones of Chile; therefore, branch canker and dieback are highly influenced by management and age conditions; hence, any factor that leads to diminished vigor may increase the incidence of canker dieback caused by Botryosphaeriaceae pathogens under our conditions. However, location cannot be totally discarded because disease development could be associated with abiotic stressors such as winter frost. Moreover, considering the information collected, previous stressor conditions in an avocado orchard, such as abiotic stress (i.e., frost damage, water stress, or root hypoxia by overirrigation) or aging management (i.e., plant growth regulator application, girdling), may increase the incidence of branch canker and dieback in avocado orchards.

This research will allow us to guide growers regarding the pathogenic conditions that are required for the development of these diseases and to perform appropriate and efficient management practices to avoid these diseases.

**Supplementary Materials:** The following supporting information can be downloaded at: https://www.mdpi.com/article/10.3390/horticulturae8121121/s1, Table S1. Data analysis of information collected for 76 variables from 16 orchards in two growing seasons (2014 and 2015) included in this study; Figure S1: Map of sixteen "Hass" avocado orchards in study. All sites corresponded to irrigated orchards of the main Chilean avocado production, located in an area of 428.97 km between Illapel (31°37′ S) and Peumo (34°24′ S); Figure S2: Biplot of the principal component analysis of predisposing factors associated with the incidence of branch canker and dieback in Chilean orchards prospected in the 2014 season. Figure S3: Biplot of the principal component analysis of predisposing factors associated with the incidence of branch canker and dieback in Chilean orchards prospected in the 2015 season.

**Author Contributions:** Conceptualization, A.L.V., P.M.G., J.S.-T., I.M.R., J.M., A.L. and A.R.; methodology, A.L.V., P.M.G. and J.S.-T.; software, A.L.V. and J.S.-T.; validation, A.L.V., P.M.G., J.S.-T., I.M.R., J.M., A.L. and A.R.; formal analysis, A.L.V., P.M.G., J.S.-T., I.M.R., J.M., A.L. and A.R.; investigation, A.L.V., P.M.G. and J.S.-T.; resources, A.L.V., P.M.G. and J.S.-T.; data curation, A.L.V., P.M.G. and J.S.-T.; writing—original draft preparation, A.L.V., J.S.-T. and P.M.G.; writing—review and editing, A.L.V., J.S.-T. and P.M.G.; visualization, A.L.V., P.M.G. and J.S.-T.; supervision, P.M.G. and J.S.-T.; project administration, A.L.V. and P.M.G.; funding acquisition, A.L.V., P.M.G., I.M.R. and J.S.-T. All authors have read and agreed to the published version of the manuscript.

**Funding:** This study was funded by Doctoral Scholarship CONICYT-PCHA 21140282, Project CONICYT-PAI 781413002, Subsole S.A., the Water and Irrigation Laboratory of Facultad de Agronomía e Ingeniería Forestal of Pontificia Universidad Católica de Chile (FAIF-UC), and Dirección de Investigación y Postgrados of FAIF-UC.

**Data Availability Statement:** Not applicable.

**Acknowledgments:** We thank the growers and advisers who allowed us to examine their orchards and the laboratory staff of our university for their technical assistance during the development of this research.

**Conflicts of Interest:** The authors declare no conflict of interest.

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
