# Peer review of "Unveiling the Predisposing Factors for the Development of Branch Canker and Dieback in Avocado: A Case of Study in Chilean Orchards"

_horticulturae, doi:10.3390/horticulturae8121121_

Round 1

Author Response

Reviewer 1 “Major revisions in multivariate analysis”

R1 Comment 1: The aim of the authors was to relate severity of branch cancer and die back, as dependent variable, to numerous variables that are expected to influence the severity of the diseases. In that sense, the use of principal component analysis is inappropriate, because it is based on correlation between independent variables, which SCD in this study is not. PCA could be based on independent variables for visual representation of relations between orchards and/or relation between independent variables regarding their highest loadings (or redundances) with selected (example by Keiser’s role, by which eigenvalue of selected principal component should be higher than 1) and rotated (for example by Varimax method, that maximizes variance of loadings of original variables with a principal component) principal components. On the other side, in spite lack of clarity in sections Methods and material and Results, it seems that applied discriminant analysis do treat severity of the disease as dependent variable (by defining classes according to the CD incidence), and the others as independent.

Authors Response: About this comments related to Principal Components Analysis (PCA), we must emphasize that in lines 232 to 235 of the revised version (lines 219-222 in the old version)  we present the NIPALS algorithm used to compute all multivariate analysis (PCA and PLS-DA). This methodologic approach (so-called Latent Projection in literature) allows computing a group of multivariate methods like PCA, Regression PLS (Partial Least Square), Discriminant Analysis (PLS-DA), and others.

NIPALS (Nonlinear Iterative Partial Least Squares) was developed by Herman Wold in 1966 and modified by Svante Wold in 1978 and perfectioned by Martens and Wold in 1980, like a simple but efficient algorithm to estimate the parameters of many multivariate techniques. The algorithm allows to use no-normal multivariate data, a large number of correlated variables (collinearity), missing data, and ill-conditioned data, that is, an incomplete rank matrix. So, is not necessary to fulfill the basic assumptions from “classic Factorial Analysis”. We suggest checking the articles of Wold et.al. (2001); Ferrer (2007) and Ericksson et.al. (2013) (32, 33, and 34 in our references).

R1 Comment 2: However, by transforming one quantitative parameter to a quantitative one to form classes led to a certain loss of information. More appropriate methods are multiple regression analysis and stepwise regression analysis which are capable to treat CD incidence as a quantitative parameter and dependent variable.

Authors Response: The conformation of classes is the basis of Discriminant Analysis using NIPALS algorithm and does not imply loss of variance, hence it is possible to apply to our data. So, we can conform groups with a common centroid each one (Fisher’s primary concepts) based on similarities suggested by PCA formally by Contribution Plots of score data (data not shown, but cited in the text), then submit to hypothesis test (PLS Discriminant Analysis) to maximize the distance among the different classes’s centroid.

On the other hand, multiple regression analysis computed by Ordinary Least Square (OLS) is not possible to apply, given is not possible to fulfill the basic assumptions of that method (the collinearity, missing data, and incomplete rank of our datasets). Therefore, is preferable to use NIPALS algorithm.

R1 Comment 3: The applied multivariate statistical methods are methods of parametric statistics i.e. the normality of variables is obligatory in order to these methods be applied. However, many variables have not normal distribution of frequencies like variables that are: ordinal, with clear ordering of the categories (like IP, Distribution, SCD), or categorical, without intrinsic ordering, with two (like PasteP, FungicideP, Ca+, HumicA (for that, by the way, both No and Yes got the same statistics), Frost) or more categories (like Zone, Rstock, Isystem, DateP, FrP, GRsite, Grdate, Pests, Diseases).

Response: We repeat our previous commentary: The NIPALS algorithm allows to use no-normal multivariate data, a large number of correlated variables, (collinearity), missing data, and ill-conditioned data, that is, incomplete rank matrix. So, is not necessary to fulfill the basic assumptions from “classic Factorial Analysis”

R1 Comment 4: Coding of the latest group is the most problematic since the coding criterion is not clear. Thus, I suggest methods of nonparametric statistics to be used. Otherwise, parametric statistical methods could be applied only with variables that have normal distribution of frequencies or those that are transformed to achieve normal distribution of frequencies. For example, CD incidence, or percentage of plants that sustained intensity of damage that could be considered to have economic importance, as variables that present partition in the population, can be transformed by arcsine transformation. Also, categorical variables (like Zone, Rstock, Isystem, DateP, FrP, GRsite, etc.) could be suggested to be used as factors in univariate factorial analysis of variance.

Authors Responses: The use of Dummy Variables is widely used in the literature; thus, we must introduce a factor that has two or more distinct levels to calculate a “continuized” variable (Stahle & Wold, 1987; Draper & Smith, 1998).

Stahle & Wold (1987). Multivariate Data Analysis and Experimental Design in Biomedical Research. In: Progress in Medicinal Chemistry - Vol. 25, edited by G.P. Ellis and G.B. West 1988, Elsevier Science Publishers. Draper & Smith, (1998). “Dummy” Variables. In: Applied Regression Analysis, 3th Ed., Cap. 14, John Wiley & Sons, Inc.

Reviewer 1 “Revision in text”

Material and Methods

  • Line 160. Total solar irradiance (TSI) instead of total solar radiation (RAD)

Response: We prefer the abbreviation to remain as it is, since changing it implies making changes in figures and tables, and there is also an explanation of the abbreviation in the text.

  • Line 229-231. It should be clarified how categorical variables like Zone, Rstock, Isystem, DateP, FrP, GRsite, etc, were transformed into ordinal.

Response: The procedure was cited in the text. We include Table S1 in this line to indicate the categories for each variable (see line 245 in the revised version of the Manuscript).

  • Lines 249. The classes should be defined.

Response: the classes were defined in the results (in the revised version please see lines 353-356 (scenario 1), 379-384 (scenario 2), and 407-412 (scenario 3)).

Results

  • Line 307. I would suggest that R2VX be represented in text by its name: redundancy (it’s square root is loading i.e. Pearson’s correlation coefficient between original variable and principal component).

Response: R2VX: Explained Variance for each variable in matrix X (dataset)

  • Line 305. How many PCA variables were selected and by which criteria?

Response: concerning the validation criteria, NIPALS Algorithm uses a full Cross-Validation routine (see lines 232-235).

  • Line 305 and Line 311. Is there difference between PCA variable and Factor, and if is not the same name should be used.

Response: We have not denoted a difference between PCA variable and Factors. That is because of these terms are used widely in the literature

  • Line 307 and Line 311. Is there difference between independent variance and total variance, and if is not the same name should be used.

Response: the individual fractional variables variance has been detailed like R2VX (see lines 323-324 in the revised version of the Manuscript), and on the other hand the total variance expained by extracted factor (I.E. PCA) is denoted like "total variance" (see line 327)

  • Line 326-330. Letters a and b in labels of orchards should be explained.

Response: Explanations and letters were included in Figures 3, 4, 5, 6, S2 and S3.

  • Line 330 “Factors=2” is redundant.

Response: Suggestion accepted

  • Comments on PCA stand for all scenarios: It would be useful to present association of selected Factors with CD incidence, as a way to evaluate the importance of influence on CD incidence for original variables related to selected Factors.

Response: In lines, 307 to 311 the is the explanation for those relation. Since PCA is a descriptive kind of model, we explain the CD incidence with an Inferential Method (PLS-DA).

  • Lines 334-336. Classes were “previously defined”, as it is stated In line 334, so they cannot be “identified” (line 336). It has been not clearly explained in section Material and methods by which criterion the classes were defined. It seems to me that classes were defined by the authors according to CD incidence. In that way, it is redundant to explain that the classes were characterized by certain levels of CD incidence as something that had been found after analysis. Clarify.

Response: The suggestion was accepted, and “identified” was eliminated from these lines. (see line 353)

  • Lines 336-339. The original variables are not discussed properly. They should be related to Factors regarding loadings with them and in that way discussed their influence on differences between orchards in CD incidence.

Response: This analysis was indicated in the results (see lines 307-412) and discussions (see lines 449-569).

  • Comments on PLS-DA stand for all scenarios. The study lacks relation between scenarios. I this case the cophenetic correlation coefficient (between distances among orchards, based on scores from PCA or PLS-DA, in two scenarios could be used.

Response: We repeat our previous commentary: The NIPALS algorithm allows to use no-normal multivariate data, a large number of correlated variables, (collinearity), missing data, and ill-conditioned data, that is, incomplete rank matrix. So, is not necessary to fulfill the basic assumptions from “classic Factorial Analysis”

Conclusions

  • Lines 575-576. Clarify

Response: Suggestion accepted. See revised Manuscript

  • Lines 575-580. Sentence too long and could be divided and clarified.

Response: Suggestion accepted. See revised Manuscript

Supplementary

  • Line 600 - Table S1. There is no value coded for SCD – healthy trees

Response: We appreciate the suggestion because there was an error when adding the data. (see Table S1, SCD – healthy trees)

  • Lines 604-607. I would suggest that either figures S2 and S3 be included in text or figures 5 and 6 be put in Supplementary

Response: We prefer the original order for the figures indicated.

  • Lines 628-649. Appendix A is redundant or incorporated in Material and methods in more compressed version and in the context of explaining the reason of the implementation of PCA.

Response: Suggestion accepted. A comprised version was included in the Statistical analysis section of Materials and methods (see lines 218-230).

Reviewer 2 Report

This a good investigation for the branch canker and dieback occurred on Chilean avocado, and the authors obtained enough data and results, which can be used as the guide to efficiently manage the practices to avoid these diseases. however, there are some points need to be improved for the writing and presentation as following suggestions:

1. please impove the whole text in writing. For examples, Line 3:title:“case of study”should be “a case of study”;Line 21:“our study”should be “our findings”;Line 25:“abiotic stress”should be “abiotic stresses”;Line 27:“In addition”should be “Moreover". please check the whole text.

2.  Table 2 showed many fungi species in Botryosphaeriaceae ,it is better to provide some pictures for these fungi morphology and their evolutionary relationships.

3. The figures need to be improved: (1) the fonts are too smaller (see Fig. 2); (2) the fonts are superimposed on the other fonts or the legends(see Figs 3,4 and 5).

Author Response

Reviewer 2

  • Please improve the whole text in writing. For examples, Line 3:title:“case of study”should be “a case of study”;Line 21:“our study”should be “our findings”;Line 25:“abiotic stress”should be “abiotic stresses”;Line 27:“In addition”should be “Moreover". please check the whole text.

Response: Suggestions accepted, please read lines 3, 21, 25, and 27.

  • Table 2 showed many fungi species in Botryosphaeriaceae ,it is better to provide some pictures for these fungi morphology and their evolutionary relationships.

Response: Adding these pictures is not relevant to the subject of this article. In this case, the most important thing is the incidence of the different species reported.

  • The figures need to be improved: (1) the fonts are too smaller (see Fig. 2)

Response: Suggestion accepted (see line 321)

  • The fonts are superimposed on the other fonts or the legends(see Figs 3,4 and 5).

Response: overlapping between observations and between variables occurs by approximation in the dimensional hyperplane. To clarify was included the names of variables in the Figure description.

Round 2

Reviewer 1 Report

please check the attachment

Author Response

Reviewer 1 “The work still needs major revision regarding implemented statistical analysis”

R1 Comment 1: My general remark was that the use of PCA in this work to relate with severity of branch cancer and die back, as dependent variable, to numerous variables that are expected to influence the severity of the diseases is inappropriate, because it is based on correlation between independent variables, which SCD and Icd in this study are not, as it is claimed in the title. PCA could be based on independent variables for visual representation of relations between orchards and/or relation between independent variables. So, the authors need to perform another statistical method that treats SCD or Icd as dependent variables (or one of them). Discriminant analysis do treat severity of the disease as dependent variable (by defining classes according to the CD incidence), and the others as independent. However, even in this case Icd was treated as independent variable, while the title claims different. Maybe the following reference could be useful:

Vicente-Gonzalez, L., Luis Vicente-Villardon J. (2022): Partial Least Squares Regression for Binary Responses and Its Associated Biplot Representation. Mathematics 2022, 10, 2580.

https://doi.org/10.3390/math10152580

Author’s Response: We appreciate Reviewer 1 observations. However, as we have stated in our previous clarifications, we must reiterate that we use of Principal Components Analysis (PCA) as a descriptive method of variability explanation, i.e., search the maximum variances least square projection of a single X matrix (not the correlation matrix). Hence, we search for the simple relationships among the group of variables. Thus, PCA is a valid and widely applicable method.

All our computing is based on the robust NIPALS algorithm (Nonlinear Iterative Partial Least Squares) was developed by Herman Wold in 1966 and modified by Svante Wold in 1978 and perfectioned by Martens and Wold in 1980, like a simple but efficient algorithm estimating the parameters of many multivariate techniques. The algorithm allows us to use no-normal multivariate data, a large number of correlated variables (collinearity), missing data, and ill-conditioned data, that is, an incomplete rank matrix. So, is not necessary to fulfill with the basic assumptions from “classic Factorial Analysis”. The objective is the simple description of the variance of the variables (no correlation), all of them assumed like equivalents and independent without no previous assumption.

Is important to remember that PCA is one of the most popular multivariate techniques because it reduces the dimensionality, compresses the noise, and relates measurements in a simple informational sub-space of the data set (Brereton, 2009).

In your last letter, you enclose an example paper related to Partial Least Square Regression (PLS), based on “a modification of NIPALS algorithm that is equivalent to PLS-2” (according to authors). The great difference is the basis of the PLS method: maximize the covariance between an X matrix (independent variables) concerning to Y matrix (1 or more dependent variables). Is important to emphasize that our aim using PCA to describe variability, hence we don’t group the variables in dependent or independent conditions.

Formally, PLS is a linear multivariate regression method whereby the multivariate variables correspond to the observations. This modeling technique establishes the relationship between two sets of predictors (X matrix) and response variables (Y matrix). PLS is a correlation analysis that estimates the values of one (or more) variable(s) from a set of controllable independent variables (Eriksson et.al., 2013).

On another hand, PLS-DA is a supervised linear discrimination method, i.e., the researcher defines previously the classes to conform, generally suggested by description method like PCA.

We enclose some own published papers using the same methodology without no objections, using an analogous methodology approach. Please revise the articles:

Ferreyra, R., Sellés, G., Saavedra, J., Ortiz, J., Zúñiga, C., Troncoso, C., Rivera., S.A., González-Agüero, M., Defilippi, B.G. 2016. Identification of pre-harvest factors that affect fatty acid profiles of avocado fruit (Persea americana Mill) cv. ‘Hass’ at harvest. South African Journal of Botany 104: 15–20.

Saavedra, J., Córdova, A., Gálvez, L., Quezada, C., Navarro, R., 2013. Principal Component Analysis as an exploration tool for kinetic modeling of food quality: A case study of a dried apple cluster snack. Journal of Food Engineering 119: 229-235.

Toledo M.S., Armijo, P., Godoy, L., Saavedra, J., Ganga, M.A. 2018. Determination of Effects of Genetic Diversity of Oenococcus oeni and Physicochemical Characteristics on Malolactic Fermentation Across Chilean Vineyards, using Multivariate Methods. Journal of Pure and Applied Microbiology 12(1): 15-21.

Pino, C., Sepúlveda, B., Tapia, F., Saavedra, J., García-González D.L., Romero, N. 2022. The Impact of Mild Frost Occurring at Different Harvesting Times on the Volatile and Phenolic Composition of Virgin Olive Oil. Antioxidants 2022, 11, 852. https://doi.org/10.3390/antiox11050852

Rivera, S., Ferreyra, R., Robledo, P., Sellés, G., Arpaia, ML., Saavedra, J., Defilippi, B. 2017. Identification of preharvest factors determining postharvest ripening behaviors in ‘Hass’ avocado under long-term storage. Scientia Horticulturae 216: 29-37.

Hence, we are sure to be using robust, reliable, efficient, and extensively tested and validated methods by the current mainstream of DataScience.

R1 Comment 2: Total solar irradiance (TSI) and instead of total solar radiation (RAD) are not the same parameters, and the authors have to state correctly which parameter they measured and used in the study.

Author’s Response: We appreciate the author´s observation. The correct term for RAD is Average Solar Radiation, which in fact is the data collected and represented in all tables and figures. We made changes in all the Manuscript and Table legends in order to correct that definition.

R1 Comment 3: The study lacks a relation between scenarios, so the importance of studying the differences between them is not clear.

Author’s Response: We include a new sentence in the paragraph that explain the use of 2 seasons. Therefore, the paragraph now indicates:

“During each growing season (2014 and 2015), commercial Hass avocado orchards were characterized, collecting information on 76 variables, corresponding to the main characteristics of each orchard, historic reports of biotic and abiotic stresses, and specific conditions or cultural management of orchards during the seasons in the study (for details please see Table S1). We include data from the two previous consecutive seasons to expand the data set of each variable, in order of considering the recent history of the orchard prior to the sampling and identification of CD.” Please see lines 148-154.

Round 3

Reviewer 1 Report

See attachments
